# Sustainable Utilization of Pulp and Paper Wastewater

**Xiaoli Liang** [1,2], **Yanpeng Xu** [1,2], **Liang Yin** [3], **Ruiming Wang** [1,2], **Piwu Li** [1,2], **Junqing Wang** [1,2] and **Kaiquan Liu** [1,2,*]

1    State Key Laboratory of Biobased Material and Green Papermaking (LBMP), Qilu University of Technology (Shandong Academy of Sciences), Jinan 250353, China; 10431211135@stu.qlu.edu.cn (X.L.); 10431211110@stu.qlu.edu.cn (Y.X.); wrm@qlu.edu.cn (R.W.); piwuli@qlu.edu.cn (P.L.); wangjq@qlu.edu.cn (J.W.)
2    Key Laboratory of Shandong Microbial Engineering, College of Bioengineering, Qilu University of Technology (Shandong Academy of Sciences), Jinan 250353, China
3    Laboratory of Microalgae Genetic Engineering, Gansu Engineering Technology Research Center for Microalgae, Hexi University, Zhangye 734000, China; yinl03@163.com
*    Correspondence: liukq@qlu.edu.cn

**Abstract:** The pulp and paper industry plays an important role in the global economy and is inextricably linked to human life. Due to its large scale, the production process generates a large amount of wastewater, which poses a major threat to the environment. The sustainable utilization and safe treatment of pulp and paper wastewater can effectively reduce environmental pollution, improve resource utilization efficiency, protect water resources, provide economic benefits for pulp and paper enterprises, and thus promote the green and sustainable development of the pulp and paper industry. Therefore, this study discusses the pollution components of pulp and paper wastewater and their impact on the environment and human health. In this review, we aim to explore the sustainable development of pulp and paper wastewater by summarizing the characteristics of current pulp and paper wastewater, the commonly used treatment methods for pulp and paper wastewater, the application of pulp and paper wastewater recycling, and the future development direction of pulp and paper wastewater.

**Keywords:** pulp; paper; wastewater treatment; sustainable utilization

## 1. Introduction

Pulp and paper is an industrial process in which finely divided raw materials such as wood, straw, or other plant fibers are used to produce paper via various processes such as crushing, soaking, grinding, fiber separation, and coagulation [1]. The pulp and paper industry is one of the most important industrial sectors in the world, and its global output value in 2020 reached USD 580 billion. People's lives are closely related to the pulp and paper industry. Global apparent consumption of paper and paperboard in 2021 is 428.51 million tons, up 6.6% from 401.81 million tons in 2020. Global apparent consumption per capita was 55.1 kg. North America has the highest apparent consumption per capita at 197.6 kg, followed by Europe and Oceania at 116.8 kg and 96.0 kg, respectively. Per capita apparent consumption is 48.5 kg in Asia, 45.1 kg in Latin America, and only 7.2 kg in Africa [2]. An overview of the pulp and paper industry capacity in 2022 in seven countries, including Australia, Brazil, and Canada, is shown in Table 1 [3]. According to relevant statistics, the total output of paper and paperboard in the global pulp and paper industry was ~417 million tons in 2022, of which the total output of the United States was ~65.95 million tons and that of China reached ~124.32 million tons [4]. Being the largest developing country in the world, China ranks first in the total amount of paper products in the world; however, its per capita output of paper products is considerably lower than that of the developed countries. Thus, China's pulp and paper industry has great potential for development. According to statistics, China's pulp, paper, and paper products industry achieved a total output of 283.91 million tons of pulp, paper, paperboard, and paper products in 2022, which reflects an annual

increase of 1.32%. In that year, the paper and paperboard output was 124.25 million tons, which reflects an increase of 2.64% over the previous year. The pulp output was 85.87 million tons, which reflects an increase of 5.01% over the previous year, and the paper products output was 73.79 million tons, a decrease of 4.65% over the previous year [4]. The industry's operating income was USD 218.7 billion, indicating an increase of 0.44% over the previous year; the total profit was USD 8.9 billion, an annual growth of −29.79%. In 2022, there were ~2500 paper and paperboard manufacturers in China, and the production volume of paper and paperboard in China was 124.25 million tons, a 2.64% increase over the previous year. The consumption was 124.03 million tons, which reflects an increase of −1.94% over the previous year, and the per capita annual consumption was 87.84 kg [4]. From 2013 to 2022, the average annual growth rate of China's paper and paperboard production was 1.87% and that of consumption was 2.59%.

**Table 1.** Overview of pulp and paper industry capacity in various countries in 2022.

| | Total Pulp Production (1000 Tones Air) | | | | |
|---|---|---|---|---|---|
| Country | Wood Pulp for Paper and Paperboard | Pulp of Other Fibre for Paper and Paperboard | Dissolving Pulp, Wood + Other Raw Materials | Paper and Paperboard | Utilization of Recovered Paper for Making Paper and Paperboard |
| Australia | 923 | 0 | 0 | 3024 | 1691 |
| Brazil | 24,969 | - | 670 | 11,040 | 5090 |
| Canada | 13,600 | - | - | 8600 | 2850 |
| China | 21,150 | 5580 | - | 124,320 | 66,420 |
| America | 40,822 | - | - | 65,959 | 29,054 |
| Japan | 7579 | 4 | 155 | 13,661 | 15,947 |
| Korea | 277 | - | - | 11,254 | 8315 |
| Thailand | 1082 | 151 | 89 | 5374 | 5201 |

The development of the pulp and paper industry is an important economic activity. However, wastewater generated during related processes is an important source of environmental pollution [5,6]. Wastewater pollutants produced by the pulp and paper industry include organic and inorganic substances. Organic substances include lignin, starch, amylase, and cellulose, whereas inorganic substances primarily include sulfates, chlorides, and nitrates [7]. In recent years, the treatment and sustainable use of pulp and paper wastewater has become an important environmental concern over the globe [8].

Since the development of the pulp and paper industry, different pulp methods such as mechanical, chemical, chemical–mechanical, and biomechanical pulp have been developed to meet different paper needs. At present, the chemical pulp method is extensively employed in the industry, and primarily includes alkaline, kraft, sulfite, organic solvent, and mechanochemical pulp [9,10]. Chemical pulp involves the separation of lignin from fibrous material to obtain cellulose by exposing papermaking materials to chemicals. This method is highly productive and extensively employed in the paper industry, accounting for ~75% production of the total industrial and commercial paper in the market [11]. However, this method produces a large amount of wastewater. It is also one of the main sources of wastewater in the pulp and paper industry. Pulp and paper wastewater includes wastewater from pulp and from different stages of paper. The composition of wastewater is considerably complex [5,12]. Figure 1 shows the sources of wastewater from the pulp and paper process.

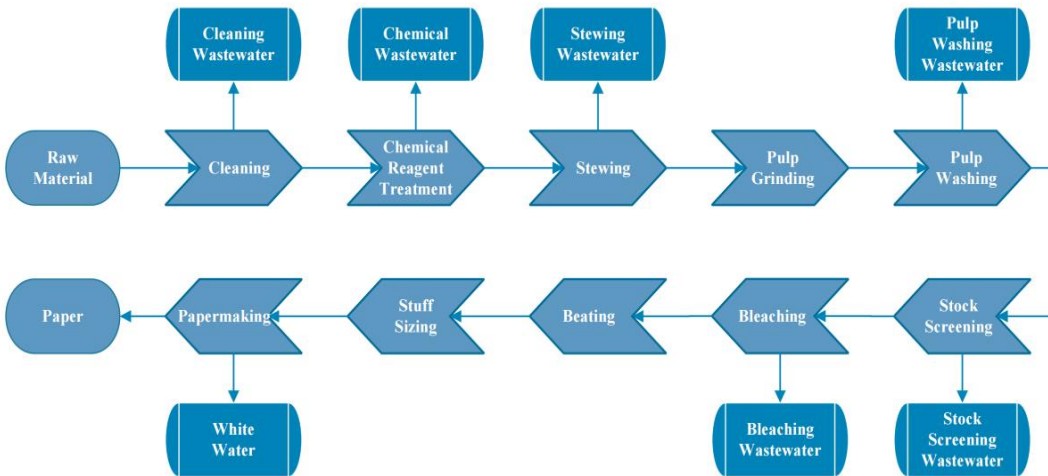

**Figure 1.** Sources of wastewater from the pulp and paper process.

The pulp process involves disintegrating paper raw materials, separating most of the fibers, separating the remaining pulp by adding various chemical paper agents to ensure good quality of the paper, and bleaching. At this stage, a large amount of high-concentration pulp wastewater is produced [13].

Black liquor is mainly generated from wastewater produced during cooking, which is the most polluting wastewater in the entire paper process. It contains a large amount of organic matter, a high concentration of chemical oxygen demand (COD), and high chroma along with a large amount of refractory fibers and inorganic salts [14]. Wastewater in the middle stage of paper primarily refers to all wastewater except black liquor, including washing, bleaching, and paper wastewater discharged during the paper process [15]. Wastewater from the washing and screening process is similar to black liquor; however, its concentration is not as high as that of black liquor [1,16]. The paper bleaching process separates the remaining fiber pigments. Due to the addition of a large amount of chlorine-containing bleaching agent, the discharged wastewater contains a large amount of chlorine-containing organic matter, which has high chroma and toxicity [17]. During the paper process, the pulp is transformed into paper. Wastewater discharged from this process is also called white water and contains various fibers and rubber materials, fillers, and preservatives that were added during the paper process. Compared with black liquor, the pollution caused by white water is minor, i.e., white water is slightly less toxic than bleaching wastewater [18].

Low-molecular-weight compounds (chlorophenols and other organochlorine compounds) in pulp wastewater discharged during the pulp process are the main cause of biomutagenicity and bioaccumulation in rivers due to their hydrophobicity and ability to penetrate cell membrane sources [19]. The toxic effects of absorbable organic halide range from carcinogenicity and mutagenicity to acute toxicity. Their accumulation and slow degradation can lead to high COD. Research has shown that the discharge of pulp and paper wastewater not only increases the toxicity caused to phytoplankton and zooplankton due to increasing biochemical oxygen demand, COD, and other pollution parameters but also induces the growth of fecal coliform bacteria, which poses a health hazard to these organisms [20]. Approximately half of the chemicals that can affect the endocrine system are chlorides (such as dioxins and polychlorinated biphenyls); studies have confirmed that these chlorinated hydrocarbon organic pollutants are primarily produced by their chemical structural changes due to lignin involved in the cooking and bleaching processes [12]. When chemical substances that are similar in structure to biological hormones enter the human body, they destroy the balance of human hormones, resulting in endocrine disorders and, in severe cases, abnormal development and reproductive functions [21]. The US Environmental Protection Agency (EPA) surveyed some paper mills and found that

60% of wastewater discharged from the paper mills contained dioxin components [22]. Dioxin exhibits the characteristics of teratogenicity, carcinogenicity, and mutagenicity [23]. It not only affects the secretion of hormones that feminize males and influence the growth and development of children but can also causes multiple brain neuropathy and acute toxicity [24]. Its toxicity is equivalent to more than 1000 times that of potassium cyanide. As a polymer compound, lignin has some molecular structures similar to the basic structure of dioxins, which are very easy to produce when pulp is bleached using chlorine [25]. Thus, the most effective method for minimizing dioxin emissions from pulp mills is limiting the use of chlorine-containing compounds [26]. Pulp and paper wastewater also contains some nonpolar compounds, such as halogenated hydrocarbons and polychlorinated biphenyls, making their water solubility considerably low [5]. In pulp and paper wastewater, chloroform, pentachlorophenol, chlorinated vanilla aldehydes, chlorinated guaiacol, and other substances exist [27]. Such toxic substances are not easily degraded biochemically or non-biochemically, and their discharge into natural water bodies has a toxic effect on organisms. The substances have strong toxicity and carcinogenic effects; at low concentrations, they chronically accumulate to cause lesions, and at high concentrations, they cause death [28]. They are enriched through the food chain or directly affect mammals and humans through drinking water [29]. Consequently, some countries have established limits for the discharge of absorbable organic halides per ton of the pulp from the paper industry [10].

With the development of society, people now have increased awareness of environmental protection and pay increased attention to sewage generated by various industries. Various sewage treatment methods have been developed and applied accordingly [1]. At present, technologies for the sustainable use of pulp and paper wastewater primarily include sewage treatment, biological treatment, recycling, and comprehensive utilization [30]. Among them, sewage treatment is the most commonly used technology at present; its purpose is to reduce pollutants in wastewater. Organic matter in sewage is primarily treated by biological treatment, whereas inorganic matter is primarily treated via chemical treatment to reduce pollutants [23]. Biological treatment is based on the microbial analysis of organic matter in wastewater. Its advantage is high treatment efficiency, and the disadvantage is high energy consumption [31]. Wastewater reuse technology involves reusing wastewater during the pulp and paper process. The advantage of this technology is that it can effectively reduce the discharge of wastewater; however, its disadvantage is that it requires advanced treatment of organic and inorganic substances in wastewater to meet the discharge standards [32]. Comprehensive utilization involves converting the pollutants in wastewater into useful substances. The advantage of this technology is that it can effectively utilize pollutants in wastewater; the disadvantage is that the related technology is complex and expensive [33].

When paper wastewater is discharged without proper treatment, it can alter the natural water body's environment from aerobic to anaerobic, disrupt the acid–base balance, diminish the water body's self-purification capabilities, and ultimately lead to water quality deterioration. Thus, the effective treatment of pulp and paper wastewater is of utmost importance. Recycling valuable substances within this wastewater and implementing industrial wastewater recycling within or between enterprises can not only safeguard water environments and ecological balance but also conserve water resources, reduce pollutants, and promote sustainable development. This effort has important implications for mitigating water resource shortages, reducing the influx of toxic substances into the ecological environment, minimizing water production costs for businesses, advancing economic and societal development, and attaining "carbon reduction" objectives. Achieving sustainable use of pulp and paper wastewater is a multifaceted challenge that requires comprehensive consideration of technical, economic, and environmental factors to identify optimal solutions [6].

We aim to explore the sustainable development of pulp and paper wastewater by summarizing the characteristics of current pulp and paper wastewater, the commonly

used treatment methods for pulp and paper wastewater, the application of pulp and paper wastewater recycling, and the future development direction of pulp and paper wastewater!

## 2. Literature Search

The scientific documents related to this review are mainly from the CNKI database, the WOS database and the FAOLEX database.

In the process of database retrieval, several combinations of the following keywords are applied: pulp; paper; wastewater treatment; sustainable utilization. All references are original articles published before October 2023. We removed duplicates and selected by title, keyword, summary, relevance. A total of 95 publications covering pulp and paper making, pulp and paper wastewater treatment, and wastewater treatment for further processing and utilization met our inclusion criteria. We acknowledge that some documents may have been omitted; however, we believe that the full range of studies collected faithfully represents the current state of knowledge on the subject.

Our results consist of three main parts. The first part mainly introduces the characteristics of pulp and paper wastewater, this part quotes 15 references. The second part mainly introduces common treatment methods for pulp and paper wastewater, 25 references are cited in this part. In this paper, the advantages and disadvantages of common methods used for treating pulp and paper wastewater are compared by table, and corresponding references are provided. The third part mainly introduces the utilization of pulp and paper wastewater. In this part, 16 references are quoted to introduce the relevant contents of the recycling of pulp and paper wastewater.

## 3. Characteristics of Pulp and Paper Wastewater

Wastewater generated by the pulp and paper industry is characterized based on chemical pollution and microbial pollution [34].

### 3.1. Chemical Pollution Characteristics

Pulp and paper wastewater is a high-concentration multicomponent pollutant. Its main constituents include suspended solids, carbohydrates, ammonia nitrogen, cyanide, sulfate, phosphate, heavy metals, and the different concentration ranges are shown in Table 2 [35].

**Table 2.** Different concentration ranges of chemical pollutants.

| Pollutant | Content (mg/L) | Test Method |
|---|---|---|
| Suspended solids | >500 mg/L | Gravimetric determination; Direct microscopic count |
| Carbohydrates | 100~500 mg/L | Combustion oxidation nondispersive infrared absorption method |
| Ammonia nitrogen | 50~200 mg/L | UV spectrophotometry |
| Cyanide | 5~20 mg/L | UV spectrophotometry |
| Sulfate compound | 100~500 mg/L | Barium chromate spectrophotometry |
| Phosphate | >20 mg/L | Ammonium molybdate spectrophotometry |

Suspended solids: pulp and paper wastewater contains a large amount of suspended solids, such as pulp debris, cellulose, and lignin. The content of these suspended solids can reach more than 500 mg/L [36]. Lignin is the most abundant of these suspended solids, accounting for >90% of the total suspended matter in pulp and paper wastewater [37].

Carbohydrates: Carbohydrates are another major pollutant in pulp and paper wastewater, which mainly include sugars, acetic acids, and carboxylic acids, and are primarily derived from cellulose, hemicellulose, and other substances decomposed during the paper process. Furthermore, some additives, such as bleaching and auxiliary agents, are used in the paper process, which leave a certain amount of carbohydrates in wastewater [38]. The carbohydrate content in pulp and paper wastewater is generally between 100 and 500 mg/L.

Ammonia nitrogen: Ammonia nitrogen is an important pollutant in pulp and paper wastewater, and its content lies generally between 50 and 200 mg/L. The content is mainly derived from added preservatives, pulp agents, lye, and other products. Pulp and paper wastewater contains ammonia and nitrogen compounds such as ammonia, amino acids, and amino sugars. The concentration of these nitrogen-containing substances can exceed 50 mg/L [39].

Cyanide: Cyanide is an important pollutant in pulp and paper wastewater. Its content typically ranges between 5 and 20 mg/L. It is primarily generated from the addition of bleaching agents and additives [40].

Sulfate compound: Sulfuric acid is a common reagent used to adjust the acidity during pulp and paper production because of its low price. Aluminum sulfate (namely, alum and bauxite) exhibits pH regulation, flocculation, media, and other effects and is also commonly used in pulp and paper production. The use of sulfuric acid and aluminum sulfate increases the concentration of sulfate in sewage [41]. Sulfate compound content in pulp and paper wastewater is generally between 100 and 500 mg/L.

Phosphate: Pulp and paper wastewater contains phosphate substances such as calcium and sodium phosphate. The content of these phosphor sources can reach >20 mg/L [10].

The chemical pollutants in pulp and paper wastewater are primarily suspended matter, carbohydrates, ammonia nitrogen, cyanide, sulfate, and phosphate compounds, among which the content of organic and suspended matter is high and can reach tens to hundreds of specific gravity [42]. In addition, other substances, such as metal ions and dyes, may impact water quality. Therefore, to effectively control chemical pollution by pulp and paper wastewater, effective control measures must be undertaken, such as reducing the use of raw materials, improving related technologies and wastewater treatment processes, and adopting effective purification technologies [43].

*3.2. Microbial Pollution Characteristics*

Pulp and paper wastewater is a type of pollutant that contains a large amount of organic and inorganic matter and microorganisms. Microbial pollution characteristics refer to the polluting biological substances present in pulp and paper wastewater as well as the type, quantity, and activity of these biological substances [40].

Bacteria, fungi, and viruses are the most common microorganisms found in pulp and paper wastewater, with bacteria being the most abundant, followed by fungi, and viruses being the least abundant. Bacteria in pulp and paper wastewater primarily include *Bacillus*, *Proteus*, anaerobic bacteria, *Escherichia aerogenes*, *Escherichia coli*, *Pseudomonas aeruginosa*, *Staphylococcus aureus*, *Klebsiella pneumoniae*, and *Vibrio cholerae* et al. [44]. These bacteria are highly active and can decompose organic matter to produce pollutants.

In addition to bacteria, pulp and paper wastewater contains several fungi, such as yeast. Yeast can generate a large amount of heat through reproduction, thereby affecting the temperature of the wastewater; it can also metabolize harmful toxins, causing pollution of the environment [45]. Further, the wastewater contains several other fungi, primarily including various molds and mold spores. These fungi can produce a large amount of organic acids, reduce the pH level of the wastewater, and produce various toxic substances, thereby polluting the environment [46].

Pulp and paper wastewater may contain many viruses, which can enter the human body through wastewater, causing infectious diseases and health hazards [47].

The microbial pollution characteristics of pulp and paper wastewater primarily refer to microorganisms such as bacteria, yeast, fungi, and viruses contained in it as well as their types, quantities, and activities. These microorganisms can produce several pollutants and cause infectious diseases, posing health hazards [48].

## 4. Common Treatment Methods for Pulp and Paper Wastewater

### 4.1. Physical Treatment

Physical treatment refers to the use of physical methods to remove pollutants from pulp and paper wastewater. Commonly used physical treatment methods for pulp and paper wastewater include sedimentation, filtration, centrifugation, flotation, and membrane filtration [49]. Sedimentation is a simple and economical method for removing suspended solids from wastewater. The principle of sedimentation is to use gravity to separate suspended matter from the wastewater. Adding flocculants or coagulants to the wastewater can improve settling processes [50]. In the flocculation precipitation method, a flocculant forms coagulation agents that are used to remove suspended particles, colloidal macromolecules, and small pollutants from water through various mechanisms. Currently, this method is gaining popularity for the tertiary treatment of pulp and paper wastewater. Kim et al. used polyaluminum chloride (PAC) as a coagulant and cationic polyacrylamide (PAM) as a flocculant over a pH range of 2–10 to treat paper wastewater through the coagulation–flocculation process. The optimal conditions for coagulation–flocculation were found to be as follows: PAC concentration of 3689 mg/L, c-PAM concentration of 39.9 mg/L, and pH of 5.4. This suggests that coagulation–flocculation may be an effective pretreatment process for paper wastewater prior to the biological treatment [51]. Filtration is the process of removing suspended solids from wastewater using filtering media. Commonly used filtering media are sand, gravel, and activated carbon. The addition of flocculants or coagulants to wastewater can improve the filtration process [52]. Centrifugation is the process of separating suspended solids from wastewater using centrifugal force. The flotation process uses air bubbles to separate suspended solids from wastewater. Further, the membrane filtration process uses membranes to separate suspended solids from wastewater. Physical treatment can effectively remove suspended solids and organic matter from wastewater; however, the removal effect on toxic substances is unsatisfactory. A specialized film is used to separate the solute from the solvent, enabling the solute in the solution to permeate through the film and accomplish the separation of the solvent. Tonni et al. conducted an investigation into the feasibility of membrane filtration processes, including reverse osmosis (RO), for treating white water generated during thermal mechanical pulp in pulp and paper mills. The study revealed that membrane treatment could effectively eliminate $SiO_2$ from wastewater samples. When subjected to initial concentrations of 200 mg/L and pressures of 11 and 10 bar, ultrafiltration or nanofiltration membranes could be used for 12 h to remove 80% and 85% of $SiO_2$, respectively. RO, under the same conditions, almost entirely eliminated targeted contaminants at 10 bar. The treated wastewater produced through membrane filtration complied with local legislation standards, and the wastewater contents are less than 50 mg [53,54]. The advantages of physical treatment methods are high treatment efficiency and good treatment effect; however, there are also certain disadvantages, such as the generation of a large amount of waste liquid during the treatment process and high treatment costs [55].

### 4.2. Chemical Treatment

Chemical treatment refers to the use of chemicals to remove pollutants from wastewater. Commonly used chemical treatment methods for pulp and paper wastewater include coagulation, precipitation, oxidation, and adsorption [56]. Coagulation involves using a coagulant to destabilize and agglomerate suspended solids in wastewater and then separating them from the wastewater. Commonly used coagulants include inorganic substances such as ferric chloride and aluminum sulfate and organic substances such as polyacrylamide [57]. Sedimentation is the process of settling suspended solids in wastewater using chemicals [58]. Commonly used precipitants are lime and soda ash. Oxidation involves the use of oxidants to oxidize organic matter in wastewater, and commonly used oxidants are chlorine and ozone. In advanced oxidation disposal method, advanced oxidation reactions are used to eliminate pollutants from industrial wastewater. Strong oxidants containing highly active hydroxyl radicals are introduced into the wastewater, causing

advanced oxidation reactions between oxygen anion free radicals and organic pollutants in the wastewater. This process gradually transforms them into safe and harmless inorganic compounds, purifying the wastewater. The method has shown favorable results in treating industrial wastewater with heavy water pollution, particularly in the pulp and paper industry [40]. Adsorption involves the use of adsorbents to adsorb pollutants in wastewater, and commonly used adsorbents include activated carbon and zeolite [23,59]. Adsorption is achieved through a chemical reaction occurring between the surface of an adsorbent and contaminants in the wastewater. In the chemical adsorption process, the adsorbent's surface usually features unsaturated bonds or active functional groups, allowing for chemical reactions with wastewater pollutants, forming chemical bonds or complexes. The principle underlying wastewater treatment via adsorption is based on the high affinity of adsorbents for pollutants in the wastewater. The choice of adsorbents is closely related to the characteristics, concentration, and desired discharge standards of the pollutants in wastewater. Various adsorbents exhibit varying degrees of selectivity for different pollutants. Adsorbents usually possess a large specific surface area and pore structure, providing numerous adsorption sites, which, in turn, increase adsorption capacity and efficiency. The advantages of the adsorption method for wastewater treatment include simplicity, effectiveness against various pollutants, and ability to regenerate adsorbents. However, there are certain limitations in treating wastewater by adsorption, such as the selection of appropriate adsorbents and the challenges of regeneration. Therefore, practical applications require careful consideration of wastewater characteristics and requirements, selection of suitable adsorbents, and optimization of treatment conditions to achieve optimal results. The advantages of chemical treatment include high treatment efficiency and good treatment effect; however, there are some disadvantages, such as poor removal effect of heavy metal ions, generation of a large amount of waste liquid during the treatment process, and high treatment costs.

*4.3. Biological Treatment*

Biological treatment is classified into aerobic sludge and anaerobic biological methods based on the characteristics of the microorganisms used for the treatment.

4.3.1. Aerobic Sludge Method

The Aerobic sludge method is commonly used for treating pulp and paper wastewater. It primarily includes biofilm, biofilter, bioprecipitation, and activated sludge methods [60,61]. Further, it uses the degradation mechanism of aerobic microorganisms to degrade organic matter in wastewater into carbon dioxide, water, and inorganic salts, thereby achieving the purpose of purifying wastewater [62]. The activated sludge and biofilm methods also use microorganisms to biodegrade organic matter in wastewater [63,64]. The advantages of biological treatment include low investment costs, low operating costs, and effective treatment. However, there are some disadvantages, such as lengthy treatment processes and difficulty in degrading heavy metals and antibiotics in wastewater [65].

4.3.2. Anaerobic Biological Method

The anaerobic biological method is suitable for treating pulp and paper wastewater with medium to high levels of pollutant concentrations. A wastewater treatment reaction device—upflow anaerobic sludge blanket (UASB reactor)—is employed, which is widely used in wastewater treatment in the pulp and paper industry around the world. Wastewater is introduced into the UASB reactor's bottom as evenly as possible. As the wastewater ascends through a sludge bed comprising granular or flocculent sludge, anaerobic reactions occur upon contact with sludge particles. The biogases produced in the anaerobic state (mainly methane and carbon dioxide) promote internal circulation, which is beneficial for the formation and maintenance of granular sludge. Some of the gases produced in the sludge layer adhere to the sludge particles. These gases, both attached and unattached, rise to the top of the reactor. When the sludge rises to the surface, it encounters the bottom of

the gas emitter in the three-phase reactor. This interaction leads to the degassing of the sludge floc attached to the bubbles. Once the bubbles are released, the sludge particles settle back onto the sludge bed. Meanwhile, the attached and unattached gases are collected in the gas collecting chamber located within the three-phase separator situated at the top of the reactor. In current engineering practice, the methods of choosing an upflow anaerobic filter and anaerobic biofilter anaerobic baffle reactor are the most frequently employed [66].

### 4.4. Advanced Integrated Processing Technology

In addition to the above technologies, other technologies for the treatment and application of pulp and paper wastewater, such as electrochemical and ultrasonic treatment, have emerged in recent years [67]. Electrochemical and sonication treatments are the processes of removing pollutants from wastewater using electricity and ultrasonic waves, respectively [68]; these methods are gradually being employed in the industry.

Comprehensive treatment is the most commonly used method in the treatment of pulp and paper wastewater [69]. It primarily uses biological, chemical, and physical treatments as well as other technologies to separate harmful substances such as organic matter and heavy metals from wastewater [70]. The ozone biological method involves directing water, after undergoing ozone treatment, into a biological filter from the bottom, where aeration is provided as needed. This biological filter is equipped with an activated carbon filler as a biological carrier, allowing for the further removal of residual COD through biological contact oxidation. Cui et al. assessed the suitability of the ozone biological method for treating wastewater with low COD concentrations, such as raw water with a COD of only 50 mg/L. This method can achieve a COD level of <30 mg/L. In cases where raw water has a high COD concentration and strict discharge standards apply, a two-stage ozone biological series treatment can be performed. This approach eliminates the need for additional chemicals, ensures safety, and minimizes labor requirements. Melchiors et al. evaluated the efficiency of a physicochemical pretreatment combination, combining the coagulation–flocculation–sedimentation (CFS) process with the Fenton advanced oxidation process for treating wastewater generated from industrial chemithermal–mechanical pulp (CTMP) in Brazil. When the CFS and Fenton processes were combined for CTMP wastewater treatment, the total organic carbon removal rate reached 95%, the COD removal rate reached 61%, and the lignin removal rate reached 76% [40]. The advantages of comprehensive treatment include good treatment effects and high treatment efficiency; however, there are some disadvantages, such as high treatment costs [71]. At present, the commonly used pulp and paper wastewater treatment methods and their performance comparison are shown in Table 3.

**Table 3.** Common methods used for treating pulp and paper wastewater.

| Handling Method | Performance Comparison |
|---|---|
| Flocculation precipitation method | The method is simple, the investment is not high, the operating cost is low, but it is difficult to reach the standard [51]. |
| Advanced oxidation treatment | The method is simple, there is no sludge, the safety is poor, the investment is not high, and the operation cost is high [40]. |
| Adsorption treatment | The investment is reasonable, but the regeneration is troublesome and the operation cost is high [59]. |
| Aerobic sludge process | Less investment, low operating cost, good treatment effect. However, the treatment process is long, and it is difficult to degrade heavy metals and antibiotics in wastewater [61]. |
| Biological ozone method | High investment, simple operation, reasonable operating cost, more suitable for upgrading [70]. |
| A combination of coagulation–flocculation–precipitation process and Fenton advanced oxidation process | The operation is stable, the operation cost is low, but the operation amount is large, the safety is poor, and the sludge amount is large [45]. |

## 5. Utilization of Pulp and Paper Wastewater

At present, wastewater generated during the pulp and paper process is typically used as a resource after treatment. Common utilization technologies for pulp and paper wastewater include recycling and energy recovery [72]. Recycling refers to the process of using wastewater to produce paper or other products. Energy recovery is the process of using wastewater to generate electricity or heat.

### 5.1. Water Recycling

Pulp and paper wastewater has agricultural applications; treated pulp and paper wastewater can be used for agricultural irrigation. Organic matter, as well as nitrogen, phosphorus, and potassium in pulp and paper wastewater can provide nutrients to the plants, improve soil fertility, and promote crop growth [73]. Pulp and paper wastewater also has industrial applications; treated pulp and paper wastewater can be used for industrial purposes, which can reduce freshwater consumption as well as replace freshwater usage [74]. Wastewater from a pulp and paper workshop at Shandong Century Sunshine Paper Group was comprehensively treated. The process involved flocculation, primary anaerobic treatment, secondary aerobic treatment, coagulation, and precipitation. Some of the treated wastewater was further treated by sand filtration and subsequently used for the greening of the paper workshop and factory area. Excess wastewater that remains after treatment is discharged [75]. Treated pulp and paper wastewater can be used to power sewage treatment plants, wash equipment, clean streets, and other purposes [76].

### 5.2. Energy Recovery

Biogas produced in the anaerobic reaction treatment of pulp and paper wastewater is a good bioenergy resource. The biogas produced is used for power generation or as a substitute for coal and other energy sources, ensuring self-sufficiency in the wastewater treatment system. Any excess electrical energy is transmitted for use in the pulp and paper process. This approach achieves energy saving and reduction in energy consumption during wastewater treatment, ultimately leading to decreases carbon emissions. During the anaerobic reaction process, ~75% of COD is converted into methane gas. Gas production at 35 °C is calculated to be 0.4 $m^3$ of biogas per 1 kg of COD removed. The biogas output of an anaerobic reactor is at least 6500 $m^3$ per day [77]. Biogas produced during the anaerobic treatment process is collected and discharged via torch combustion, which pollutes the environment and wastes energy. Hence, the biogas is purified, and methane gas purified through biological desulfurization is used for hot air drying of paper machines, power generation, and factory heating [78]. Shandong Century Sunshine Paper Group employs this method for black liquor treatment. Ali et al. showed that using paper mill biosludge to produce biochar and bioenergy leads to substantial reduction of carbon emissions. The advantages of using paper mill biosludge as biochar, extend beyond mitigating climate change. The benefits include enhanced nutrient use efficiency and soil fertility, particularly in forests with poor soil quality, as well as improved soil acidification and soil health through heavy metal management [47].

### 5.3. Alkali Recovery

Many pulp and paper manufacturers perform alkali recovery because pulp and paper wastewater contains a high concentration of alkali. Alkali recovery technologies typically include combustion causticization, electrodialysis, and hydrothermal carbonization [79]. Among them, the combustion causticization method involves four production processes: extraction, evaporation, combustion, and causticization calcium oxide recovery. The combustion causticization method can only recover the alkali used in pulp, and a large amount of lignin is burned after black liquor is evaporated to dryness. The method involves high energy consumption and requires multiple pieces of evaporation equipment, making small- and medium-sized pulp mills unsuitable for the implementation of this method [80]. The electrodialysis method involves passing ions in black liquor from a low-concentration

region to reach a high-concentration region through a semipermeable membrane. Under the action of a DC electric field, alkali can be recovered on the anode side. Further, the pH value of the cathode side is reduced due to the enrichment of hydrogen ions, and some lignin is precipitated; however, the recovery rate of this method is only 50% [81]. The hydrothermal carbonization method uses carbon dioxide as the acidifying agent, and organic impurities in black liquor are precipitated as carbonization raw materials in a closed environment. Compared with the traditional alkali recovery method, this method has low energy consumption, no wastewater and residue discharge, and can produce high-value-added activated carbon products [82]. Ji et al. demonstrated the feasibility of the direct causticization technology using reed black liquor with limestone as the bedding material (Figure 2). NaOH was successfully recovered through direct causticization. Approximately 91.2% of the sodium introduced into the system could be effectively recovered, with 87.4% of it converted into NaOH [83].

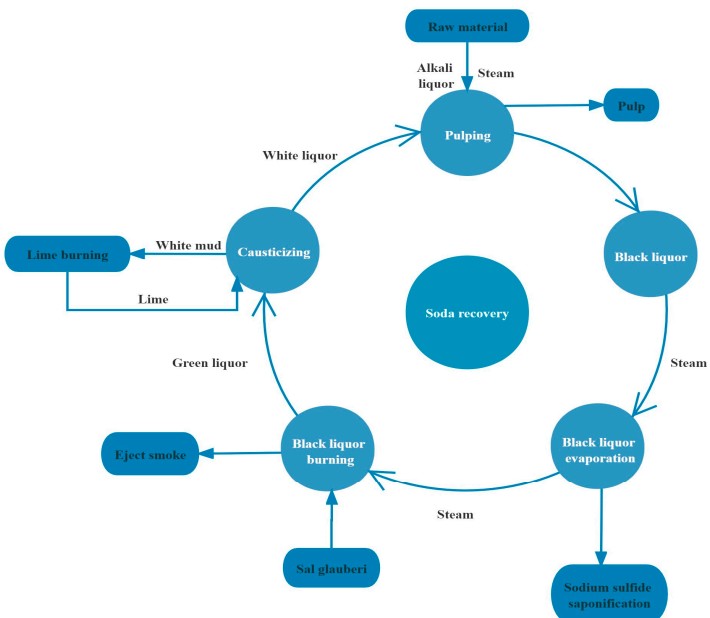

**Figure 2.** Flow chart of alkali recovery process in black liquor.

### 5.4. Lignin Recovery

The organic components in black liquor produced during the pulp process mainly include lignin and polysaccharides. During the cooking process, due to the action of sodium hydroxide, the ether bonds in lignin break and combine with hydroxide ions to form alkali lignin [12]. Alkali lignin is a natural polymer surfactant. After chemical modification, it can be used as a nontoxic renewable resource to replace petrochemical products; it can also be used as an adsorbent for heavy metal treatment, which has high application value. Alkali lignin can be precipitated by lowering the pH value. Common methods include adding acid to black liquor or introducing carbon dioxide. Jeroen et al. developed a pilot-scale procedure for mild alkaline pulp, followed by the recovery of lignin from black liquor. In a pilot-scale experiment, lignin recovery was achieved through acidification, enzyme treatment, and subsequent filtration. The study revealed that the purity of lignin was higher when a flocculant was employed [84]. The optimal condition for acid precipitation and dealkalization of lignin include a pH of <3, and the precipitation rate of up to 90% or higher can be achieved at a temperature of 25 °C. Figure 3 shows the lignin recovery process of the acid analysis of black liquor [85].

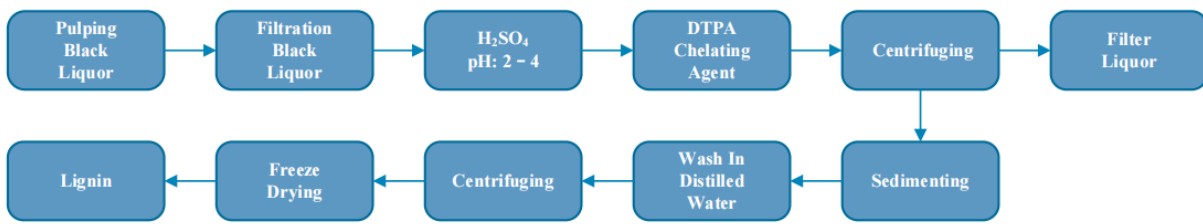

**Figure 3.** Lignin recovery process of the acid analysis of black liquor.

## 6. Sustainable Utilization of Pulp and Paper Wastewater

Sustainable utilization refers to minimizing environmental pollution, protecting natural resources, and realizing sustainable resource use in the case of limited resources and energy [86,87]. In the paper process, a large amount of wastewater is produced, which contains wood pulp, chemical additives, pigments, and other harmful substances. Effectively using this wastewater to reduce environmental pollution is an important concern in paper manufacturing [88].

First, technological transformation must be sought. Investment in technological transformation should be increased, and new technologies should be promoted and adopted, such as activated sludge treatment, wet dust removal, and ion exchange technology, to effectively reduce pollutant emissions [89]. It is also necessary to improve technical guidance for enterprises, raise environmental protection awareness among enterprises, encourage enterprises to improve technology, and effectively control wastewater discharge [89,90].

Further, the funding guarantee mechanism must be improved. Financial support for technological transformation should be increased and provided for enterprise transformation; the cost of enterprise transformation should also be reduced [91]. Furthermore, tax incentives should be offered to enterprises to encourage them to invest in environmental protection technologies to promote the development of wastewater treatment technologies [40].

Finally, an efficient regulatory system must be established. For different types of paper wastewater, corresponding discharge standards should be established. Further, an effective monitoring and testing system should be established, and strict penalties should be imposed on enterprises with excessive discharge [92]. In addition, an effective information-sharing mechanism should be established to promptly release information about pollution sources, and regular inspections of pollution sources should be conducted to ensure that enterprises comply with emission standards [93].

To develop an effective and sustainable use policy of pulp and paper wastewater, we must strengthen technological transformation, improve the fund guarantee mechanism, and establish an efficient supervision system [88,91]. Hence, by these discussed solutions, we can effectively control the discharge of pulp and paper wastewater, protect the environment, and effectively use resources [94].

## 7. Conclusions and Future Perspectives

With the development of society, the output of paper products—the products of the pulp and paper industry—is increasing, and paper products are inextricably linked to human life. Wastewater from the pulp and paper process must be safely treated and discharged. Various methods, such as advanced oxidation, membrane filtration, and activated sludge method, can be employed for paper wastewater treatment. Each of these methods is suited for different water quality conditions, and all these methods hold valuable research potential. In recent years, domestic and international researchers have increasingly focused on the combination of multiple processes for paper wastewater treatment. Thus, there is a need to explore the combinations of chemical, physical, and biological methods to maximize wastewater treatment efficacy and achieve cost efficiency.

In the 21st century, people's awareness of environmental protection is increasing, resulting in a paradigm shift; for example, using biological enzymes in the production

process and all-biological pulp using high-efficiency biological enzymes. Recycling pulp and paper wastewater has become the future trend toward the green development of the pulp and paper industry. For example, manufacturers modify the pulp and paper process to recycle wastewater discharged during the process. If potassium hydroxide is used instead of sodium hydroxide in the pulp and paper process (there is no difference between the two in terms of pulp performance), discharged pulp and paper wastewater will contain sufficient potassium. Potassium fertilizer can be obtained from the discharged pulp and paper wastewater through secondary production, which not only offers economic benefits to factories but also reduces wastewater pollution. People created the first paper without the use of chemicals over 2000 years ago. With the advancement of modern biotechnology, highly efficient enzymes derived from organisms suitable for the pulp and paper industry have emerged. Because of their mild action conditions and high action efficiency, biological enzymes can effectively reduce energy consumption of the pulp and paper process and reduce wastewater pollution. We anticipate that by implementing new technologies and processes, the pulp and paper process will become more environmentally friendly and efficient in the future.

**Author Contributions:** K.L. and X.L. conceived the study. K.L., X.L., L.Y., Y.X. and R.W. wrote the draft of the manuscript. P.L. and J.W. critically reviewed the full manuscript content. All authors have read and agreed to the published version of the manuscript.

**Funding:** This work was supported by the National Key Research and Development Program of China (No. 2019YFC1905902); the Foundation (No. 202008) of Qilu University of Technology of Cultivating Subject for Biology and Biochemistry; the Foundation (No. 2022GH026) of International Technology Cooperation Project from Qilu University of Technology (Shandong Academy of Sciences); the Foundation (No. FWL2021065) of Shandong Provincial Key Laboratory of Biophysics; and the Foundation (No. 21YF5FA129) of Gansu Provincial Key R&D Program-Social Development. The funders had no role in study design, data collection and analysis, decision to publish, or preparation of the manuscript.

**Data Availability Statement:** The datasets supporting the results of this article are included within the article.

**Conflicts of Interest:** The authors declare no conflict of interest.

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
