# Peer review of "Sustainable Utilization of Pulp and Paper Wastewater"

_water, doi:10.3390/w15234135_

Round 1
Reviewer 1 Report
Comments and Suggestions for Authors
Overall, this article systematically outlines the characteristics, common treatment methods, and sustainable use ways of pulp and papermaking wastewater, which has good reference value. There are no major grammatical or expression errors. However, there are some aspects that could be improved:
1. The introduction could receive help from a clearer explanation of the significance of sustainable use of pulp and papermaking wastewater.
2. The description of the chemical pollution characteristics could be more specific by supplying concrete data on the concentration ranges of different pollutants, to give readers a more intuitive understanding of the wastewater pollution.
3. In the treatment technology section, consider dividing it into subsections to detail various physical, chemical, biological methods, etc., and analyze their advantages and disadvantages. Also briefly mention which methods are most widely used in current engineering applications.
4. For wastewater resource use, supplement some typical engineering application examples to help readers better understand the various use ways. Specifically describe processes, application situations, etc. of alkali recovery, lignin recovery, etc. Also, the authors could supply more specific examples of the potential applications of treated wastewater, such as the use of wastewater in agriculture or the production of biofuels.
5. The article could receive help from a more detailed discussion of the challenges and limitations of sustainable use of pulp and papermaking wastewater, such as the need for further research on the environmental impact of different treatment technologies and the potential for contamination of groundwater and surface water.
6. The article needs a clearer and more complete conclusion summarizing the core viewpoints and values of the full text, and outlook future research directions.
7. In formatting, there are errors in the numbering sequence of figures and tables that need to be renumbered. The reference format also needs to be unified.

Overall, this article systematically outlines the characteristics, common treatment methods, and sustainable use ways of pulp and papermaking wastewater, which has good reference value. There are no major grammatical or expression errors. However, there are some aspects that could be improved:
1. The introduction could receive help from a clearer explanation of the significance of sustainable use of pulp and papermaking wastewater.
2. The description of the chemical pollution characteristics could be more specific by supplying concrete data on the concentration ranges of different pollutants, to give readers a more intuitive understanding of the wastewater pollution.
3. In the treatment technology section, consider dividing it into subsections to detail various physical, chemical, biological methods, etc., and analyze their advantages and disadvantages. Also briefly mention which methods are most widely used in current engineering applications.
4. For wastewater resource use, supplement some typical engineering application examples to help readers better understand the various use ways. Specifically describe processes, application situations, etc. of alkali recovery, lignin recovery, etc. Also, the authors could supply more specific examples of the potential applications of treated wastewater, such as the use of wastewater in agriculture or the production of biofuels.
5. The article could receive help from a more detailed discussion of the challenges and limitations of sustainable use of pulp and papermaking wastewater, such as the need for further research on the environmental impact of different treatment technologies and the potential for contamination of groundwater and surface water.
6. The article needs a clearer and more complete conclusion summarizing the core viewpoints and values of the full text, and outlook future research directions.
7. In formatting, there are errors in the numbering sequence of figures and tables that need to be renumbered. The reference format also needs to be unified.
Author Response
Responses to reviewer 1:
Thank you for the comments made by the reviewer. We modified the manuscript based on the reviewer’s comments. The detailed answers to the comments are as follows:
General comments:
Overall, this article systematically outlines the characteristics, common treatment methods, and sustainable use ways of pulp and papermaking wastewater, which has good reference value. There are no major grammatical or expression errors. However, there are some aspects that could be improved.
Answer: Thank you for your suggestion. We have made corresponding changes according to your suggestions.
Detailed comments:
Question 1 The introduction could receive help from a clearer explanation of the significance of sustainable use of pulp and papermaking wastewater.
Answer: Thanks for your comment. We have re-complemented the importance of sustainable use of pulp and paper wastewater in L165-176.
Question 2 The description of the chemical pollution characteristics could be more specific by supplying concrete data on the concentration ranges of different pollutants, to give readers a more intuitive understanding of the wastewater pollution.
Answer: Thanks for your comment. We have provided specific data for different pollutant concentration ranges in Table 2 to characterize chemical pollution more specifically.
Question 3 In the treatment technology section, consider dividing it into subsections to detail various physical, chemical, biological methods, etc., and analyze their advantages and disadvantages. Also briefly mention which methods are most widely used in current engineering applications.
Answer: Thank you for your suggestion. In the third part, various physical, chemical and biological treatment methods for pulp and paper wastewater are introduced in detail, and their advantages and disadvantages are analyzed. Through Table 3, we briefly introduce the methods that are widely used in current engineering applications.
Question 4 For wastewater resource use, supplement some typical engineering application examples to help readers better understand the various use ways. Specifically describe processes, application situations, etc. of alkali recovery, lignin recovery, etc. Also, the authors could supply more specific examples of the potential applications of treated wastewater, such as the use of wastewater in agriculture or the production of biofuels.
Answer: Thanks for your comment. In the fourth part, we have re-supplemented some typical pulp and paper plant engineering application examples. Supplementary Figure 2 describes in detail the process flow of alkali recovery in pulping and paper black liquor. The application of alkali recovery and lignin recovery were introduced in detail. Provide more concrete examples of the potential applications of treated wastewater in, for example, biofuels.
Question 5 The article could receive help from a more detailed discussion of the challenges and limitations of sustainable use of pulp and papermaking wastewater, such as the need for further research on the environmental impact of different treatment technologies and the potential for contamination of groundwater and surface water.
Answer: Thanks for your comment. In the third part, we have re-supplemented the advantages and disadvantages of different pulping and paper wastewater treatment methods, and have been concerned about the possibility of subsequent pollution.
Question 6 The article needs a clearer and more complete conclusion summarizing the core viewpoints and values of the full text, and outlook future research directions.
Answer: Thanks for your comment. We have revised a clearer and more complete conclusion, summarizing the core points and values of the paper, and looking forward to future research directions.
Question 7 In formatting, there are errors in the numbering sequence of figures and tables that need to be renumbered. The reference format also needs to be unified.
Answer: Thanks for your comment. We have readjusted the format, re-numbered the charts and tables, and adjusted the references.

Reviewer 2 Report
Comments and Suggestions for Authors
Upon careful review, it became apparent that the content presented in the article did not offer significant contributions to the existing body of knowledge on the subject matter. The ideas presented were not sufficiently innovative and did not provide fresh perspectives on the sustainable utilization of pulp and papermaking wastewater. While I appreciate the effort put into the research, I regret to inform you that I found the article lacking in novelty, coherence, and organization.
Comments on the Quality of English LanguageA certain degree of editing is necessary to refine the English language used in this context. Some moderate adjustments and revisions are needed to enhance the clarity, coherence, and overall quality of the text.
Author Response
Responses to reviewer 2:
Thank you for the comments made by the reviewer. We modified the manuscript based on the reviewer’s comments. The detailed answers to the comments are as follows:
General comments:
Upon careful review, it became apparent that the content presented in the article did not offer significant contributions to the existing body of knowledge on the subject matter. The ideas presented were not sufficiently innovative and did not provide fresh perspectives on the sustainable utilization of pulp and papermaking wastewater. While I appreciate the effort put into the research, I regret to inform you that I found the article lacking in novelty, coherence, and organization.
Answer: Thank you for your suggestion. In this paper, the pollution composition of pulp and paper wastewater and its harm to the environment and human are discussed. Different wastewater treatment methods, sustainable utilization of pulp and paper wastewater and its future development trend are summarized. Our resubmitted document discusses the future development direction of pulp and paper wastewater treatment technology in more detail. We have tried our best to improve the manuscript and sincerely hope to get your approval.

Reviewer 3 Report
Comments and Suggestions for Authors
The paper titled " Sustainable Utilization of Pulp and Papermaking Wastewater" is a timely and relevant contribution to the field of sustainable paper production and environmental protection. The study provides valuable insights into the characteristics of pulping and papermaking wastewater, the technologies employed for wastewater treatment, and potential applications of treated wastewater.
The paper addresses a critical issue within the paper industry, highlighting the importance of sustainable wastewater management in a sector that has a significant environmental footprint. The study offers an encompassing overview of the subject, discussing characteristics, treatment technologies, and potential applications of pulping and papermaking wastewater. This approach provides a holistic understanding of the topic. By discussing the applications of treated wastewater, the paper underscores the potential for resource recovery and circular economy practices in the industry, which is crucial for sustainability.
However, the paper lacks detailed descriptions of the methodologies used in the study. It is essential to provide clarity on the research methods for the benefit of readers and to facilitate reproducibility. While it mentions future trends briefly, a more in-depth exploration of potential future research directions would enhance the paper's value.
One important improvement that the paper would benefit from is referencing relevant TAPPI standards and Best Available Techniques (BAT) documents. To ensure the paper aligns with industry standards and best practices, the authors should consider referencing relevant TAPPI standards, such as those related to water, waste, and environmental management. Additionally, the paper could benefit from citing BAT references, particularly those specific to the pulp and paper industry. Incorporating these references could provide readers with practical guidance and industry standards.
Another method to enhance the paper, I recommended that the authors consider including detailed methodologies for data collection, treatment processes, and analysis of wastewater characteristics. This will provide transparency and help readers understand the reliability of the results.
Furthermore, the paper should delve deeper into future research directions. It might consider investigating emerging treatment technologies, exploring potential industrial partnerships for sustainable wastewater management, and assessing the economic viability of wastewater treatment processes.
In conclusion, this paper holds promise in addressing the pressing issue of sustainable wastewater management in the pulp and paper industry. By enhancing methodological details, discussing future research directions, and including references to official standards, the paper can further contribute to the knowledge and practice of sustainable paper production.
Comments on the Quality of English Language
none
Author Response
Responses to reviewer 3:
Thank you for the comments made by the reviewer. We modified the manuscript based on the reviewer’s comments. The detailed answers to the comments are as follows:
General comments:
The paper titled "Sustainable Utilization of Pulp and Papermaking Wastewater" is a timely and relevant contribution to the field of sustainable paper production and environmental protection. The study provides valuable insights into the characteristics of pulping and papermaking wastewater, the technologies employed for wastewater treatment, and potential applications of treated wastewater.
The paper addresses a critical issue within the paper industry, highlighting the importance of sustainable wastewater management in a sector that has a significant environmental footprint. The study offers an encompassing overview of the subject, discussing characteristics, treatment technologies, and potential applications of pulping and papermaking wastewater. This approach provides a holistic understanding of the topic. By discussing the applications of treated wastewater, the paper underscores the potential for resource recovery and circular economy practices in the industry, which is crucial for sustainability.
Answer: Thank you for your suggestion. We have made corresponding changes according to your suggestions. We made some revisions to the manuscript and marked the revised document in red.
Detailed comments:
Question 1 However, the paper lacks detailed descriptions of the methodologies used in the study. It is essential to provide clarity on the research methods for the benefit of readers and to facilitate reproducibility. While it mentions future trends briefly, a more in-depth exploration of potential future research directions would enhance the paper's value.
Answer: Thank you for your suggestion. We have made corresponding changes according to your suggestions. We have introduced the relevant methods used in this paper in more detail, providing readers with a clearer research method and more in-depth exploration of potential future research directions for pulp and paper wastewater treatment.
Question 2 One important improvement that the paper would benefit from is referencing relevant TAPPI standards and Best Available Techniques (BAT) documents. To ensure the paper aligns with industry standards and best practices, the authors should consider referencing relevant TAPPI standards, such as those related to water, waste, and environmental management. Additionally, the paper could benefit from citing BAT references, particularly those specific to the pulp and paper industry. Incorporating these references could provide readers with practical guidance and industry standards.
Answer: Thank you for your suggestion. In the third part of this paper, the industry standard of pulp and paper wastewater treatment and the current application examples of pulp and paper mills are quoted to provide practical guidance and industry standards for readers.
Question 3 Another method to enhance the paper, I recommended that the authors consider including detailed methodologies for data collection, treatment processes, and analysis of wastewater characteristics. This will provide transparency and help readers understand the reliability of the results.
Answer: Thank you for your suggestion. Table 1 is added to describe the production capacity of the pulp and paper industry in each country in 2022. New Table 2 describes the concentration range of chemical contaminants in pulp and paper wastewater. The new Table 3 briefly describes the wastewater treatment methods commonly used in the plant. New Figure 2 describes the alkali recovery process of pulping and papermaking black liquor. This paper describes the characteristics of pulp and paper wastewater in more detail in the second part and the different treatment methods in more detail in the third part, which will provide transparency and help the reader understand the reliability of the results.
Question 4 Furthermore, the paper should delve deeper into future research directions. It might consider investigating emerging treatment technologies, exploring potential industrial partnerships for sustainable wastewater management, and assessing the economic viability of wastewater treatment processes.
Answer: Thank you for your suggestion. In the sixth part of this paper, the future research direction of pulp and paper wastewater treatment is further discussed. In the fourth part of this paper, emerging treatment technologies are investigated to assess the economic feasibility of wastewater treatment processes.
Question 5 In conclusion, this paper holds promise in addressing the pressing issue of sustainable wastewater management in the pulp and paper industry. By enhancing methodological details, discussing future research directions, and including references to official standards, the paper can further contribute to the knowledge and practice of sustainable paper production.
Answer: Thank you for your suggestion. This paper adds methodological details, discusses future research directions, and includes references to official standards that could further advance the knowledge and practice of sustainable paper production.
